



# Light-dependent calcification in Red Sea giant clam *Tridacna maxima*

Susann Rossbach[1], Vincent Saderne[1], Andrea Anton[1], Carlos M. Duarte[1]

[1] Biological and Environmental Science and Engineering Division, Red Sea Research Centre (RSRC) and Computational Bioscience Research Center (CBRC), King Abdullah University of Science and Technology (KAUST), Thuwal, Kingdom of Saudi Arabia

*Correspondence to*: Susann Rossbach (susann.rossbach@kaust.edu.sa)

**Abstract.** Tropical giant clams of the Tridacninae family, including the species *Tridacna maxima*, are unique among bivalves as they live in a symbiotic relationship with unicellular algae and generally function as net photoautotrophic. Light is therefore crucial for these species to thrive. Here we examine the light-dependency of calcification rates of *T. maxima* in the central Red Sea as well as the patterns of its abundance with depth in the field. Red Sea *T. maxima* show highest densities in a depth of 3 m with $0.82 \pm 0.21$ and $0.11 \pm 0.03$ individuals $m^{-2}$ (mean ± SE) at sheltered and exposed sites, respectively. Experimental assessment of net calcification ($\mu mol\ CaCO_3\ cm^{-2}\ h^{-1}$) and gross primary production ($\mu mol\ O_2\ cm^{-2}\ h^{-1}$) under seven light levels (1061, 959, 561, 530, 358, 244 and 197 $\mu mol\ quanta\ m^{-2}\ s^{-1}$) showed net calcification rates to be significantly enhanced under light intensities corresponding to a water depth of 4 m ($0.65 \pm 0.03\ \mu mol\ CaCO_3\ cm^{-2}\ h^{-1}$; mean ± SE), while gross primary production was $2.06 \pm 0.24\ \mu mol\ O_2\ cm^{-2}\ h^{-1}$ (mean ± SE). We found a quadratic relationship between net calcification and tissue dry-mass (DM in gram), with clams of an intermediate size (about 15 g DM), showing the highest calcification. Our results show that the Red Sea giant clam *T. maxima* stands out among bivalves as a remarkable calcifier, displaying calcification rates comparable to other tropical photosymbiotic reef organism, such as corals.





## 1 Introduction

Giant clams (Family Cardiidae, Subfamily Tridacninae) are among the largest and fastest growing bivalves on earth, reaching up to one meter in size (Rosewater, 1965) and growth rates of up to $8 - 12$ cm yr$^{-1}$ in the largest species, *Tridacna gigas* (Beckvar, 1981). In the Indo-Pacific, giant clams are considered ecosystem engineering species (Neo et al., 2015), playing

multiple roles in the framework of coral reef communities, such as providing food for a number of predators and scavengers (Alcazar, 1986), shelter for commensal organisms (De Grave, 1999) and substrate for epibionts (Vicentuan-Cabaitan et al., 2014). By producing calcium carbonate shell material they can occasionally even form reef-like structures (Andréfouët et al., 2005). However, due to their specific habitat preference (Yonge, 1975;Hart et al., 1998) and their presumed longevity (Chambers, 2007) Tridacninae are exceedingly vulnerable to exploitation and environmental degradation (Ashworth et al.,

2004;Van Wynsberge et al., 2016). In Southeast Asia, giant clams are harvested for human consumption (adductor muscle and mantle meat) and for their shells (Lucas, 1994), already since pre-historic times (Hviding, 1993). Giant clams are also reared in aquaculture farms for the fishkeeping market (Bell et al., 1997), and in an effort of restocking the natural population (Gomez and Mingoa-Licuanan, 2006). Currently, all giant clam species are listed in the IUCN Red List of Threatened Species (IUCN, 2016) and protected under Appendix II of the Convention on International Trade in Endangered Species of Wild Fauna and

Flora (CITES), however most of them under a lower risk / conservation dependent status (Neo et al., 2015). Besides the pressure of fishing on natural stocks, giant clams are also predicted to be vulnerable to the effects of climate change, including heat waves which have been associated with mass die-off events of Tridacninae in French Polynesia (Andréfouët et al., 2013).

      Giant clams are one of the few molluscan groups living in symbiotic relationship with dinoflagellates of the genus *Symbiodinium* (Yonge, 1936;Taylor, 1969;LaJeunesse et al., 2018), likewise to as corals and sea anemones. They are generally

described as being mixotrophic (Klumpp et al., 1992), obtaining their energy both from filter-feeding and photosynthesis, however some species appear to be even functionally autotrophic (Beckvar, 1981;Jantzen et al., 2008). This dual capacity is assumed to support their fast calcification and growth rates, exceeding those of most other bivalves (Klumpp and Griffiths, 1994). Thus, the availability of light seems to be a critical factor affecting the growth and overall performance of giant clams (Lucas et al., 1989). To date, several studies have examined long-term growth rates of giant clams in response to different

environmental factors, such as nutrient enrichment (Hastie et al., 1992;Hoegh-Guldberg, 1997;Belda-Baillie et al., 1998), water temperature (Hart et al., 1998;Schwartzmann et al., 2011) and wave exposure (Hart et al., 1998). Only a few studies assessed net calcification of Tridacninae as a short-term process and how environmental factors, especially light, are influencing calcification, physiology and general metabolic rates of Tridacninae.

      A positive correlation between light and calcification has been observed in several photosynthetic calcifying

organisms, symbiotic (e.g. scleractinian corals) or not (e.g. coccolithophorids and calcifying algae) (Allemand et al., 2011). For corals, the term Light Enhanced Calcification (LEC) has been coined (Yonge, 1931), however the underlying mechanisms remain poorly understood and various hypotheses have been proposed: (1) The photosynthetic uptake of carbon dioxide by the symbionts lowers $CO_2$ levels, while increasing pH and the concentration of carbonate ions at the calcification site, which eventually could favour calcium carbonate precipitation (McConnaughey and Whelan, 1997), (2) the removal of inhibiting





substances (such as phosphates) by the symbionts during photosynthesis (Simkiss, 1964) or (3) the light-induced production of signalling molecules by the symbionts, could lead to an increase in enzymatic activity, essential for the calcification of the host (Ip et al., 2015). Only within the last years it has been possible to investigate LEC mechanisms at the molecular level (Moya et al., 2008;Bertucci et al., 2015) leading to an increasing number of publications reporting light-enhanced expression

of enzymes, such as carbonic anhydrase, supporting shell formation in giant clams (Ip et al., 2006, 2015, 2017; Hiong et al., 2017a, 2017b; Chan et al., 2018; Chew et al., 2019). There is also evidence for the light enhanced expression of genes encoding for those transporters / enzymes needed for calcification within the inner mantle and ctenidium of *Tridacna squamosa* (Hiong et al., 2017a;Hiong et al., 2017b;Ip et al., 2017;Chew et al., 2019). As both tissues are lacking the presence of symbiotic algae, it has been supposed that light could also directly affect the giant clam host. Despite recent progress in understanding LEC

processes in *Tridacninae*, much remains unknown to date. Previous studies mostly focussed on molecular processes or long-term (several months) effects of light on growth rates, assessed either as increase in shell length (Lucas et al., 1989;Adams et al., 2013) or total weight (Adams et al., 2013) and did not differentiate between different light intensities. Only a small number of studies actually reported short-term (hours to few days) effect of light on calcification. They either focused on the development of proxies (Strontium / Calcium ratio) for parameters of the daily light cycle (Sano et al., 2012) through tracer

(Strontium) incorporation or aimed to understand environmental and physiological parameters controlling daily trace element incorporation, using the total alkalinity (TA) anomaly technique (Warter et al., 2018). As growth and calcification rates in calcifying organisms are considered to be controlled by the corresponding light intensities (Barnes and Taylor, 1973) and as the penetration of light decreases with depth, so is the calcification rate expected to decrease (Goreau, 1963).

*Tridacna maxima*, the most abundant giant clam species in the Red Sea, can be found on shallow reef flats and edges,

usually shallower than 10 m, where light intensity is high due to these transparent waters of tropical, oligotrophic oceans (Van Wynsberge et al., 2016). Although tridacnid clams are one of the most dominant and charismatic molluscan taxa in the Red Sea (Zuschin et al., 2000) little is known about their ecology in this area. In addition, the majority of studies on *Tridacninae* in the region exclusively focused on the Gulf of Aqaba in the Northern Red Sea (Roa-Quiaoit, 2005;Jantzen et al., 2008;Richter et al., 2008), which represents less than 2% of the entire basin of the Red Sea (Berumen et al., 2013).

In the present study, we assessed the net calcification rates (as μmol calcium carbonate per hour ) of *T. maxima* in two short incubation experiments under seven different incident light levels (corresponding to a water depth of 0 – 14m) and in the dark, as well as photosynthetic rates at three experimental light level corresponding to the high light conditions in shallow waters (0 – 4m). Further, we assessed *in situ* abundances of *T. maxima* in different depth zones (0.5 – 11m) at a sheltered and an exposed reef in the Central Red Sea. To our knowledge, this is the first study quantifying the light-dependence of short-

term net calcification rates of tridacnid clams of the Red Sea, interrelating these rates with their abundances in the field.



## 2 Material and Methods

### 2.1 Clam abundance surveys

Abundance surveys on *T. maxima* were conducted either via snorkelling or SCUBA diving at two reefs in the eastern central Red Sea (Fig.1). The first station was Abu Shosha (22.303833 N, 39.048278 E), a small inshore reef, were abundances were

examined at the sheltered, leeward side (Southeast) of the reef, which are relatively protected from wave action and currents (Khalil et al., 2013). Additionally, abundances were assessed at a second station (20.753764 N, 39.442561 E), a fringing reef close to Almojermah, were we conducted transects at the exposed, windward side (Northwest) of the reef. At both stations, belt transects were conducted in six different depths (0.5, 1.5, 3, 5, 8 and 11 m). At the sheltered reef, a total area of 1,000 $m^2$ was covered and we conducted six transects at each depth. At the exposed reef 560 $m^2$ were covered, with three transects at

each depth. Transect lines of 25 m were deployed and all *T. maxima* specimen within 2 meters of the transect where counted (e.g., 50$m^2$ area was covered on each transect). In addition, their length (maximum anterior to posterior distance) was recorded at the sheltered reef, using a measuring tape to the nearest cm.

### 2.2 Clam incubations to obtain net calcification rates

We determined net calcification (see section 2.4 below) in *T. maxima* during two consecutive incubation experiments. During the first incubations, conducted in December 2016, we assessed net calcification of *T. maxima* under four different, moderate experimental light level, mimicking light intensities at different water depths ranging from 4 to 14 m and during a dark incubation. In November 2016, 20 specimen of *T. maxima* (shell length of 17 ± 2 cm; mean ± SD) were collected in a water depth of about 4 m at a sheltered reef site (Station 1) (Fig. 2). As *T. maxima* is often embedded in the substrate, specimens

were removed by carefully cutting their byssus with a knife. The incubations took place in December 2016 at the Coastal and Marine Resources Core Lab (CMOR) of King Abdullah University of Science and Technology (KAUST) in Thuwal, Saudi Arabia. The experimental setup consisted of ten flow-through independent LDPE (low density polyethylene) outdoor aquaria (30 L). Each aquaria contained two clams (in total 20), cleaned with a brush from epibionts prior to the experiment. Aquaria were supplied with water by gravity through an intermediate PVC (polyvinylchloride) tank of 77 L, itself receiving water

pumped from the adjacent Red Sea surface water at a flow of 0.22 $m^3$ $h^{-1}$, leading to a complete water exchange in each single aquaria every 80 minutes. To maintain ambient Red Sea surface water temperatures, all aquaria were immersed to the last top cm in a large flow-through pool of 12 $m^3$, receiving the overflowing water from the intermediate tank and the 10 experimental tanks. An Exo1 probe (YSI Incorporated, Yellow Springs, USA) was used to log water temperature and salinity at 30 min frequency. Both remained constant during the experimental period with an average temperature of 27.2 ± 0.8 °C (mean ± SD,

n = 672) and salinity of 38.4 ± 0.8 (mean ± SD, n = 672). Experimental aquaria were shaded with nets to reproduce light levels that mimicked natural conditions at different depths on the reef. We conducted short-term incubations of 6 hours (from approx. 09:30 to 15:30 mean solar time) under four different shadings and one dark incubation (at night) (n = 10), allowing 3 days acclimatization period to the clams, prior to each incubation. During the incubations, the flow-through system was turned off





in order to determine changes in seawater carbon chemistry over time as a measure for calcification processes. Photosynthetically active radiation (PAR) was recorded with a light logger (Odyssey Logger, Dataflow Systems Ltd., New Zealand) as µmol quanta m$^{-2}$ s$^{-1}$ and averaged over the incubation period, as natural light conditions fluctuated over the course of the day. Experimental light levels comprised 530, 358, 244 and 197 µmol quanta m$^{-2}$ s$^{-1}$. Using data on depth-dependent

decrease of light levels (Dishon et al., 2012), we calculated the extinction of light with water depth. The experimental irradiation levels therefore correspond to incident light conditions at about 4, 8, 12, and 14 m water depth. No additional food was provided, as natural and unfiltered seawater was flowing into the tanks.

During the subsequent incubation, conducted in April 2018, we examined net calcification and primary production of *T. maxima* under three additional experimental high light level, addressing light effects encountered in very shallow waters

(between 0 and 4 m). We collected eight specimen of *T. maxima* (shell length of 17 ± 1 cm; mean ± SD) in a water depth of about 4 m at an exposed, fringing reef close to Almojermah (Station 2) (Fig. 2). The incubation experiment was conducted on board of R/V Thuwal in a setup consisting of two big PVC flow-through tanks (350 L each), containing 9 individual PVC tanks (10 L), eight of them containing one clam each (cleaned from epibionts) and one serving as a control tank. To maintain ambient Red Sea surface water temperatures, all aquaria were immerged into the flow-through pool and water was constantly

pumped (0.36 m$^3$ h$^{-1}$), assuring a constant water exchange and movement in the individual tanks. Temperature and salinity were checked four times a day using a handheld CTD probe (CastAway-CTD, SonTek, USA). Both remained constant during the experimental period with an average temperature of 31.5 ± 0.3 °C (mean ± SD, n = 16) and salinity of 38.2 ± 0.1 (mean ± SD, n = 16). During the incubations, the individual tanks were closed airtight with see-through PVC lids and water movement was generated with battery-driven motors (Underwater motor, Playmobil, Germany). Nets were used for shading

and therefore to reproduce light levels that mimicked natural conditions at different depths on the reef. We conducted closed short-term incubations of 3 hours (from approx. 11:00 to 14:00 mean solar time) under three different shadings and one dark incubation (at night), allowing one day acclimatization to the clams prior to each incubation. Measurements of PAR intensities were identical to the first round of incubations. Experimental light levels comprised 561, 959 and 1061 µmol quanta m$^{-2}$ s$^{-1}$. The amount of light received by the highest experimental light level was identical to light received directly at the water surface

in the reef of collection at the same time of the day. Experimental irradiation levels correspond to incident light conditions at 0 m, 0.5 m and at 4 m. No additional food was provided, as raw unfiltered seawater was used.

## 2.3 Carbonate chemistry

At the start, after three and after six hours of incubation, seawater was sampled from each experimental aquaria in gas tight 100 mL borosilicate bottles (Schott Duran, Germany) and poisoned with mercury chloride, following Dickson et al. (2007).

Each sample was analysed for TA by open-cell titration with an AS-ALK2 titrator (Apollo SciTech,USA) using certified seawater reference material (CRM) (Andrew Dickson, Scripps Institution of Oceanography). During the incubations at moderate light levels (530, 358, 244 and 197 µmol quanta m$^{-2}$ s$^{-1}$), additional samples for dissolved inorganic carbon (DIC) were analysed using an AS-C3 infrared DIC analyser (Apollo SciTech, USA). Further components of the carbonate system





were calculated with the R package Seacarb (Lavigne and Gattuso, 2013) using first and second carbonate system dissociation constants of (Millero, 2010) as well as the dissociations of HF and $HSO_4^-$ (Dickson, 1990; Dickson and Goyet, 1994) respectively. Carbonate chemistry at the beginning of each incubation and in all experimental aquaria were comparable with mean ($\pm$ SD) TA of 2324 $\pm$ 83 and $\Omega_{Ara}$ of 3.44 $\pm$ 0.33 (n = 50) during the moderate light incubations and a TA of 2489 $\pm$ 38

(n = 4) during the high light incubations (Supplementary Material_S1).

### 2.4 Net calcification

Net calcification (G in µmol $CaCO_3$ h$^{-1}$) was estimated from changes in total alkalinity (TA) using the alkalinity anomaly technique (Smith and Key, 1975) using the following equation (Eq. 1):

$$G = - \frac{\Delta TA}{2} \times \frac{1}{\Delta t} \tag{1}$$

Where $\Delta TA$ is the variation of TA during the time (t) of the incubations and the factor 2 accounts for a decrease of TA by 2 equivalents per $CaCO_3$ precipitated (Zeebe and Wolf-Gladrow, 2001). Calcification rates were expressed relative to either

mantle surface area (cm$^2$) or tissue dry-mass (g). For mantle surface area, the power relationship between standard length in cm (L) and mantle area (cm$^2$) (Jantzen et al., 2008) was used to calculate mantle surface in cm$^2$. For tissue dry-mass (DM in gram) of clams, all clams were dissected, and their biomass was determined subsequently to the incubation experiment. Clams were opened by cutting the adductor muscle with a scalpel, the mantle and other tissues were separated from the shells and dried at 60 °C for 24 to 48 hours to determine tissue DM to the nearest 0.01 g.

Experimentally determined net calcification rates (µmol $CaCO_3$ h$^{-1}$) of *T. maxima* for different DM under four moderate light level (197, 244, 358 and 530 µmol quanta m$^{-2}$ s$^{-1}$) were used to create a multiple regression modelling net calcification for any given light level and DM.

### 2.5 Primary production

Primary production was assessed during the high light incubations (561, 959 and 1061 µmol quanta m$^{-2}$ s$^{-1}$), only. Therefore, oxygen (µmol L$^{-1}$) content in the incubation chambers was automatically logged (miniDOT, Precision Measurement Engineering, Inc., USA) in 15 minute intervals over the three-hour incubation period. Net photosynthesis (NPP) was calculated from the variation of oxygen concentration over time and normalized for clam mantle surface area (µmol $O_2$ cm$^{-2}$ h$^{-1}$). Dark respiration rates (R), also given in µmol $O_2$ cm$^{-2}$ h$^{-1}$ were used to calculate gross primary production (GPP) as

Eq. (2):

$$GPP = NPP + R \tag{2}$$






## 2.6 Statistical analyses

For assessing the comparisons of clam abundance at the six survey depths, an analysis of variance (ANOVA) and pairwise post-hoc Tukey analysis (Tukey HSC) were performed. A statistical model was built to explain calcification rates from the combination of PAR and clam tissue dry-mass. The model chosen was a multiple non-linear relationship built as the

5   combination of a linear dependency between PAR and calcification rates and a quadratic dependency net calcification rates and clam tissue mass. This model was selected against other concurrent models by using Akaike information criterion (AIC) (Burnham and Anderson 2003). Statistical analyses were performed using R (Foundation for Statistical Computing, Vienna, Austria, Version 3.4.2) and Statistica (Dell Software).



## 3 Results

### 3.1 Depth-dependent abundances

At the sheltered reef site, significantly highest abundances of *T. maxima* (0.82 ± 0.21 individuals m$^{-2}$; mean ± SE) were observed in a water depth of 3 m (ANOVA, p < 0.001, F = 35.6; Post-hoc Tukey test p < 0.001; Supplementary Material_S2_1),

being twice as high as in shallower waters (between 0.41 ± 0.02 and 0.44 ± 0.01 individuals m$^{-2}$ mean ± SE; at 0.5 and 1.5 m, respectively) (Fig. 2). No clams were found at the deepest survey depth of 11 m and abundances at 8 m were low with 0.04 ± 0.01 individual m$^{-2}$ (mean ± SE). Giant clams were significantly less abundant in deeper water when compared to shallow reef areas (p < 0.001 for both 0.5 and 1.5 m when compared to 8 and 11 m). In average, the density of *T. maxima* at the sheltered reef (0.5 – 11 m depth) was 0.32 ± 0.05 individuals m$^{-2}$; mean ± SE). The average size of clams was

16.6 ± 5.1 cm (mean ± SD, n = 422) and their calculated mantle surface area 140.4 ± 90.4 cm$^2$ (mean ± SD, n = 422) respectively.

At the exposed reef, abundances of *T. maxima* were overall lower with 0.04 ± 0.01 individuals m$^{-2}$ (mean ± SE), however we also found highest densities of clams at a water depth of 3 m (0.11 ± 0.03 individuals m$^{-2}$; mean ± SE)(ANOVA, p = 0.027, F = 3.813; Supplementary Material_S2_2), however they were only significantly higher than those found at 8 and

11 m (with mean ± SE of 0.02 ± 0.01 and 0.01 ± 0.01, respectively) (Post-hoc Tukey test; Electronic Supplementary Material_S2_2)(Fig. 2).

### 3.2 Net calcification and primary production

We combined observed net calcification (as the balance between calcification and dissolution) at all seven experimental incident light level and the dark incubation and identified a polynomial relationship (R$^2$ = 0.77) between net calcification

(NC, μmol CaCO$_3$ cm$^{-2}$ h$^{-1}$) and incident light (I, μmol quanta m$^{-2}$ s$^{-1}$) (Eq. 3) (Fig. 3),

$$NC = -2e^{-6} \times I^2 + 0.0019 \times I + 0.1643 \tag{3}$$

Among all light incubations, net calcification rates of *T. maxima* were highest (mean ± SE 0.65 ± 0.03 μmol CaCO$_3$ cm$^{-2}$ h$^{-1}$)

at experimental incident light levels of 530 to 561 μmol quanta m$^{-2}$ s$^{-1}$ (Fig. 3). *T. maxima* still showed positive, but low net calcification during the night (0.18 ± 0.02 μmol CaCO$_3$ cm$^{-2}$ h$^{-1}$; mean ± SE). The lowest NC rates (mean ± SE of 0.01 ± 0.01 μmol CaCO$_3$ cm$^{-2}$ h$^{-1}$) were observed at the highest incident irradiance of 1061 μmol quanta m$^{-2}$ s$^{-1}$. Overall, we observed a decline in net calcification with both, decreasing and increasing light intensities (Table 1), with polynomial regression indicating the maximum calcification (NC$_{max}$) to be reached at an incident light level of 475 μmol quanta m$^{-2}$ s$^{-1}$.

From an incident light level of 1033 μmol quanta m$^{-2}$ s$^{-1}$ on, we expect to see dissolution processes outweighing calcification (NC$_{min}$, -0.01 μmol CaCO$_3$ cm$^{-2}$ h$^{-1}$).

Gross primary production (GPP) under the high light incubations (561, 959 and 1061 μmol quanta m$^{-2}$ s$^{-1}$) showed an

identical decreasing trend with increase in incident light as observed for net calcification. At 561 μmol quanta $m^{-2}$ $s^{-1}$, GPP was highest (2.06 ± 0.24 μmol $O_2$ $cm^{-2}$ $h^{-1}$; mean ± SE) and production rates were significantly lower (ANOVA, p = 0.039, F= 4.982; Supplementary Material_S3), during the incubations at 959 and 1061 μmol quanta $m^{-2}$ $s^{-1}$ (Table 1, Fig. 3), with mean ± SE of 1.76 ± 0.28 and 0.87 ± 0.37 μmol $O_2$ $cm^{-2}$ $h^{-1}$, respectively. Two specimens died after the second highest light
5    treatment of 959 μmol quanta $m^{-2}$ $s^{-1}$.

We identified a quadratic relationship between net calcification and tissue dry-mass, with clams of an intermediate size (DM of about 15 g), showing the highest calcification rates at the four incubations at moderate light level (197, 244, 358, 530 μmol quanta $m^{-2}$ $s^{-1}$) (Fig. 4). Therefore, we combined the influence of light and dry-mass into a statistical model, explaining 77% of the variance in observed calcification rates (all parameters p < 0.05, Table 2, Fig. 5). Based on this model,
10   we identify maximum rates on clams of an intermediate size (DM of about 15 g), showing the highest calcification rates at the four light level incubations.



## 4 Discussion

### 4.1 Depth-dependent abundances

In the Red Sea, *T. maxima* shows a significant dependence of net calcification rates with incident light. This light-dependency is consistent with significantly higher abundances of this species in shallow, sunlit reef flats. Globally, densities of *T. maxima*
range between 0.1 to 0.0001 individuals m$^{-2}$ (Van Wynsberge et al., 2016), with some exceptions such as at the Ningaloo Marine Park in Western Australia with 0.86 clams m$^{-2}$ (Black et al., 2011), the Egyptian Sinai peninsula with peak values of 0.80 clams m$^{-2}$ (Roa-Quiaoit, 2005) and 0.42 clams m$^{-2}$ in Kiribati (Chambers, 2007).

In water depths between 0.5 and 11 m, we found averaged (± SD) abundances of *T. maxima* of 0.04 ± 0.01 individuals m$^{-2}$ and 0.32 ± 0.05 individual per m$^{-2}$ (mean ± SE) at an exposed and sheltered reef, respectively. Abundances at the sheltered
reef are ranking amongst the highest abundances reported world-wide, representing a 50% higher abundance than previously reported for a local reef (Bodoy, 1984) with 0.22 clams m$^{-2}$. This difference in average abundances between the two reefs observed in this study could be explained by the leeward and windward (sheltered or exposed, respectively) character of the examined sites. As reviewed by Van Wynsberge (2016), the 'reef type' can influence Tridacna abundances, as it potentially affects the water exchange (and thus water temperature and nutrient availability) as well as the exposure to waves. Similar to
Roa-Quiaoit (2005), we found that *T. maxima* abundances in the Red Sea seems to display great differences between locations, as we found significant lower numbers of giant clams at the exposed reef, with an average of 0.04 ± 0.01 individuals m$^{-2}$; (mean ± SE) in water depths between 0.5 and 11 m. Explanations for the observed differences in numbers of clams per m$^2$ at both reefs could be due to the contrasts in abiotic environmental conditions. Giant clams at the exposed reef are not only more imperilled to potentially higher wave action than at the sheltered reef site, they are possibly also exposed to higher surface
water temperatures due to the more southerly location of this reef. Mean annual temperature of the Red Sea have been shown to increase towards lower latitudes and can be as high as 33 °C in the Central and Southern Red Sea (Chaidez et al., 2017). In addition, local geomorphological features of each reef could influence the light availability of benthic habitats. Consequently, differences in the local topography could lead to different angles of incident light and shading conditions, which could result in differences between reefs even though the examined depths are identical.
Contrasting findings to previously reported *Tridacninae* abundances in the Central Red Sea could be further a result of differences in sampling depths in the respective studies, as e.g. Bodoy et al., (1984) only accounted for clams in water depth of maximum 2 m, while we assessed abundances of *T. maxima* in six different depths (0.5, 1.5, 3, 5, 8 and 11 m). Previous studies have shown that the depth of abundance surveys significantly impact the estimates (Van Wynsberge et al., 2016), even though generally, highest densities of *T. maxima* are always reported for shallow reefs (0 – 5 m) (Jantzen et al.,
2008;Andréfouët et al., 2009). This is also reflected in the results of previous studies in the Red Sea (Roa-Quiaoit, 2005) showing highest abundances of *T. maxima* in shallow water (< 3 m). However, Roa-Quiaoit (2005) accumulated abundances at all depths less than 3 m, while we differentiated even between the 0.5 m, 1.5 m and 3 m depth level and thereby found that although *T. maxima* shows the highest density at 3 m, abundances in shallower depths are significantly reduced.



Furthermore, we found only few specimens of *T. maxima* in water depths between 5 and 11 m. This finding is similar to previous studies, describing *T. maxima* as being mostly restricted to reefs shallower than 10 m, principally reef flats and edges (Van Wynsberge et al., 2016). This depth-distribution is most likely a result from a trade-off between maximizing light dependent photosynthesis while minimizing temperature stress, UV irradiation, wave exposure and / or emersion stress.

All these stressors have been previously reported to lead to massive bleaching and mass die-off events in *T. maxima* (Addessi, 2001) and preventing settlement and recruitment in the shallow waters of the reef flat (Watson et al., 2012). The average size of *T. maxima* specimens at the sheltered reef was $16.6 \pm 5.1$ cm ($\pm$ SD), similar to previous studies on this species in the Red Sea (Roa-Quiaoit, 2005), corresponding, according to the size classification by (Manu and Sone, 1995) to broodstock (i.e., sexually mature individuals) hermaphrodites. However, the number of small, juvenile specimens (< 4 cm) is potentially

underestimated, as they are extremely cryptic (Munro and Heslinga, 1983).

### 4.2 Light-dependent calcification and production in Red Sea giant clam *T. maxima*

Overall, we found significantly enhanced net calcification rates in Red Sea *T. maxima* during light incubations compared to the dark incubation. Net calcification rates also significantly increased with light intensity up to 475 μmol photons m$^{-2}$ s$^{-1}$

(incident light level corresponding to a water depth of approximately 5 m, at the same time of the day and season, when the incubations were conducted), thenceforward decrease until an eventual dominance of dissolution over calcification at approximately 1033 μmol photons m$^{-2}$ s$^{-1}$ (corresponding to light conditions received directly at the at the air-water interface in the reef of collection). Likewise to net calcification in *T. maxima*, we observed gross primary production (GPP) to be highest at intermediate light levels of around 560 μmol photons m$^{-2}$ s$^{-1}$ (corresponding to a water depth of about 4 m) and to decrease

with increasing light intensities (at 959 μmol quanta m$^{-2}$ s$^{-1}$ and 1061 μmol quanta m$^{-2}$ s$^{-1}$, corresponding to 1.5 and 0.5 m water depth, respectively). We conclude that net calcification in the Red Sea giant clam *T. maxima* is not only enhanced by light, but is likely coupled to the photosynthetic activity of their algal symbionts. Further, our results show that both, net calcification and primary production in Red Sea *T. maxima* are highest at incident light level received in water depths between 5 and 3 m at Red Sea reefs. This is especially noteworthy as these findings correlate with the observed depth-related abundances of

*T. maxima*, displaying highest densities in intermediate water depths around 3 m in the Central Red Sea. The observed irradiance optima for both, net calcification and primary production of *T. maxima* could therefore provide an explanation for the maximum in abundances in intermediate waters (3 – 5 m) and the decreasing numbers of observed clams at both, shallower and deeper reef sites.

Overall, our finding of enhanced calcification rates under light are consistent with reports on the related species

*Tridacna gigas* (Lucas et al., 1989), *Tridacna derasa* (Sano et al., 2012) and *Tridacna squamosa* (Adams et al., 2013). The mechanisms of light-enhanced calcification (LEC) have been intensely studied in zooxanthellate scleractinian corals, leading to several hypotheses proposed to explain LEC (Tambutté et al. 2011). The majority of these refer to mechanisms that are influenced by the symbiotic relationship of host and *Symbiodinium*, with the most supported hypothesis relating





photosynthetic $CO_2$ uptake by the algal symbionts to increase pH and the concentration of carbonate ions, thereby favouring calcification through the corresponding elevated saturation state for carbonate minerals (McConnaughey and Whelan, 1997).

The reliance of calcification of calcifying host organism (e.g. *T. maxima*) on their relationship with symbiotic algae could also provide an explanation for the significant decrease in net calcification rates at the highest light treatment (1061 μmol photons m$^{-2}$ s$^{-1}$). These diminished rates could be the result of photoinhibition and even photodamage of the associated *Symbiodinium* algae, when exposed to these high incident light levels. This would be also supported by the pronounced decrease in gross primary production rates at this light treatment. High incident light level, especially high level of UV radiation in shallow waters, have been previously shown to be correlated with decreased calcification rates in other marine calcifiers such as stony corals, e.g. *Porites compressa* (Kuffner, 2001).

However, recent findings for hermatypic corals also report that the contribution by the symbionts might not be the primary or sole driver for LEC, but the blue light spectrum could trigger the light sensors of the host itself, leading to higher calcification rates (Cohen et al., 2016). It is suggested that blue light photoreceptors in coral tissues of *Porites lutea* and *Acropora variabilis,* could potentially sense the light which is ultimately activating a cascade of processes involved in blue light-enhanced calcification (Cohen et al., 2016). However, our experimental light level, produced by different layers of neutral screen shading, only differed in light intensities but not in the wavelength that *T. maxima* would receive in the respective water depth. In a previous study on *Tridacna crocea*, short-term calcification rates were also reported to be strongly light-dependent (Warter et al., 2018). However, in this their experiment, Warter et al. (2018) exposed the clams not only to artificial light but also light level that were not comparable to actual conditions in the environment, as the average treatment comprised only 162 ± 7 μmol quanta m$^{-2}$ s$^{-2}$ (corresponding to a water depth of approximately 16 m in an oligotrophic ocean such as the Red Sea).

### 4.2.1 Allometric relationship between calcification and biomass

We determined a non-linear relationship between net calcification and biomass (as tissue DM) in *T. maxima*. Clams of an intermediate DM of approximately 15 g showed the highest net calcification throughout the four incubations as moderate light levels (530, 358, 244 and 197 μmol quanta m$^{-2}$ s$^{-1}$). Specimens of a smaller or higher biomass calcified less during the incubations. A similar allometric relationship has been previously described for the photosynthetic metabolic performance of the zooxanthellae in *T. maxima* (Yau and Fan, 2012). This allometric pattern is most likely due to an optimal ratio of symbionts to clam body-mass at intermediate sizes. As the clam grows, its mantle tissue increases in thickness and thus the three dimensional tubular system, bearing the utmost of symbionts (Fisher et al., 1985). However, as the mantle thickens, impinging light must penetrate through more tissue before reaching the stacked zooxanthellae (Trench et al., 1981) and there is evidence for increased shading of the symbionts in the mantles of bigger clams (Fisher et al., 1985). With further increasing size, the number of symbionts per unit clam biomass also decreases (Fisher et al., 1985;Fitt et al., 1993;Griffiths and Klumpp, 1996). In general, growth rates giant clams seem to decrease with age once they reached the threshold for maturity and become broodstock hermaphrodites (Van Wynsberge et al., 2016). Past this age, a growing portion of their energy is invested in



reproduction (Romanek and Grossman, 1989;Van Wynsberge et al., 2016), especially since there is an exponentially increase of produced egg numbers with increasing shell size (Jameson, 1976).

### 4.2.2 Comparison with other calcifiers

We compared net calcification rates of *T. maxima* with those of other benthic phototrophic and mixotrophic calcifiers (Table 3). In most calcifying organisms that live in symbiotic relationship with zooxanthellae (such as corals), metabolic rates and calcification are normalized by surface area. In contrast to corals however, which host their symbiotic algae intracellularly in their endodermal cell layer, the symbionts of Tridacninae are located in delicately branching and specialized channels within the mantle, which extend from the stomach (Trench et al. 1981; Norton et al. 1992). Although this difference makes the comparison to other calcifiers conceptually difficult, normalisation of calcification rates per mantle surface area would be also appropriate, as *Symbiodinium* cells in giant clams are mostly found in the upper 5 mm of the mantle (Ishikura et al. 1997).

The Red Sea giant clam *T. maxima* shows averaged net calcification rates of $0.47 \pm 0.03$ µmol $CaCO_3$ $cm^{-2}$ $h^{-1}$ (mean $\pm$ SE), which are comparable than those reported for hermatypic corals ($0.42 \pm 0.08$ µmol $CaCO_3$ $cm^{-2}$ $h^{-1}$; mean $\pm$ SE) and those for calcifying macroalgae ($0.43 \pm 0.38$ µmol $CaCO_3$ $cm^{-2}$ $h^{-1}$; mean $\pm$ SE). However, in comparison with averaged rates of other heterotrophic, temperate bivalve species, such as *Mytilus edulis*, *Argopecten purpuratus* and *Crassostrea gigas* with $0.08 \pm 0.07$ µmol $CaCO_3$ g $DM^{-1}$ $h^{-1}$ (mean; $\pm$ SD), calcification in *T. maxima* is about 70 times higher ($5.38 \pm 0.42$ µmol $CaCO_3$ g $DM^{-1}$ $h^{-1}$; mean $\pm$ SE) (Table 4). When compared to heterotrophic cold-water coral species, such as *Lophelia pertusa* and *Madrepora oculata*, net calcification rates of *T. maxima* are more than 7 times higher ($0.80 \pm 0.70$ µmol $CaCO_3$ g $DM^{-1}$ $h^{-1}$; mean $\pm$ SD); Table 4). Our comparative assessment of the net calcification rates of giant clams with temperate / azooanthellate species show that rates in the Red Sea *T. maxima* tested here comparable to other photosymbiotic organisms (such as corals) and calcifying algae.



## 5 Conclusion

The present study shows that net calcification and photosynthetic rates of Red Sea *T. maxima* are light-dependent, but show a maximum at intermediate irradiance, suggesting strong inhibition at the highest incident light levels received in very shallow (0 – 1.5 m) waters. This is consistent with the depth-related distribution of this species in the Red Sea, and elsewhere, which

5 showed maximum abundances in shallow (3 m), sunlit coral reefs, but a decrease in abundance from 3 m towards the surface and below. Albeit enhanced calcification is consequently beneficial for *T. maxima*, the light-dependency of both calcification and production restricts them to shallow waters, which also makes them more vulnerable to potentially harmful environmental changes, such as predicted increasing water temperatures associated to global warming (Hughes et al., 2003) as well as high levels of incident light, including high levels of UV radiation (Shick et al., 1995). The present study provides an important

10 baseline for future studies examining the impact of wavelength specific responses of calcification and metabolic rates on giant clams as well as for a better overall understanding of light enhanced calcification in Red Sea Tridacninae and their relationship with the symbiotic algae.



**Author contributions**

C.M.D, V.S., and S.R. conceptualized the research, V.S. and S.R. collected the animals and performed the abundance surveys. Experimental execution was carried out by S.R., A.A. and V.S. conducted the data curation and ran formal analyses. S.R. prepared the first draft of the manuscript and all co-authors contributed substantially to subsequent versions including the

5   final draft.

**Competing interests**

On behalf of all authors, the corresponding author states that there is no conflict of interest.

10   **Acknowledgments**

This research was funded by King Abdullah University of Science and Technology (KAUST) through base-line funding to C.M.D. and a fellowship of the Visiting Student Research Program to S.R. We thank J.L. Randle and F.I. Rossbach for assistance with field sampling and the KAUST Coastal and Marine Resources Core Lab for logistical support.



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





**Figures**

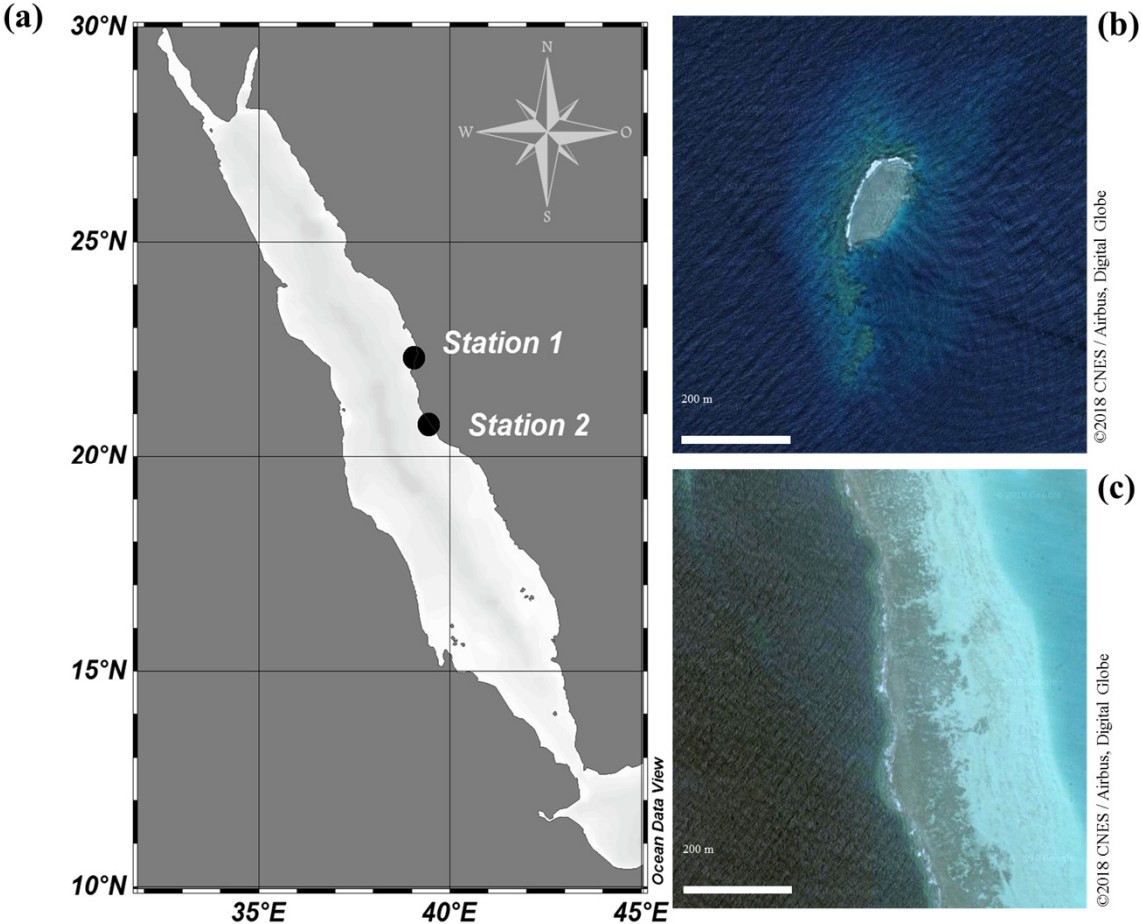

**Figure 1: (a) Map of the Red Sea. Abundance surveys and sampling of clams for incubation experiments where conducted at both, a sheltered reef (Station 1; 22.303833 N, 39.048278 E) and an exposed reef (Station 2; 20.753764 N, 39.442561 E), (b) satellite image of sheltered reef (Station 1), (c) satellite image of exposed, fringing reef (Station 2).**

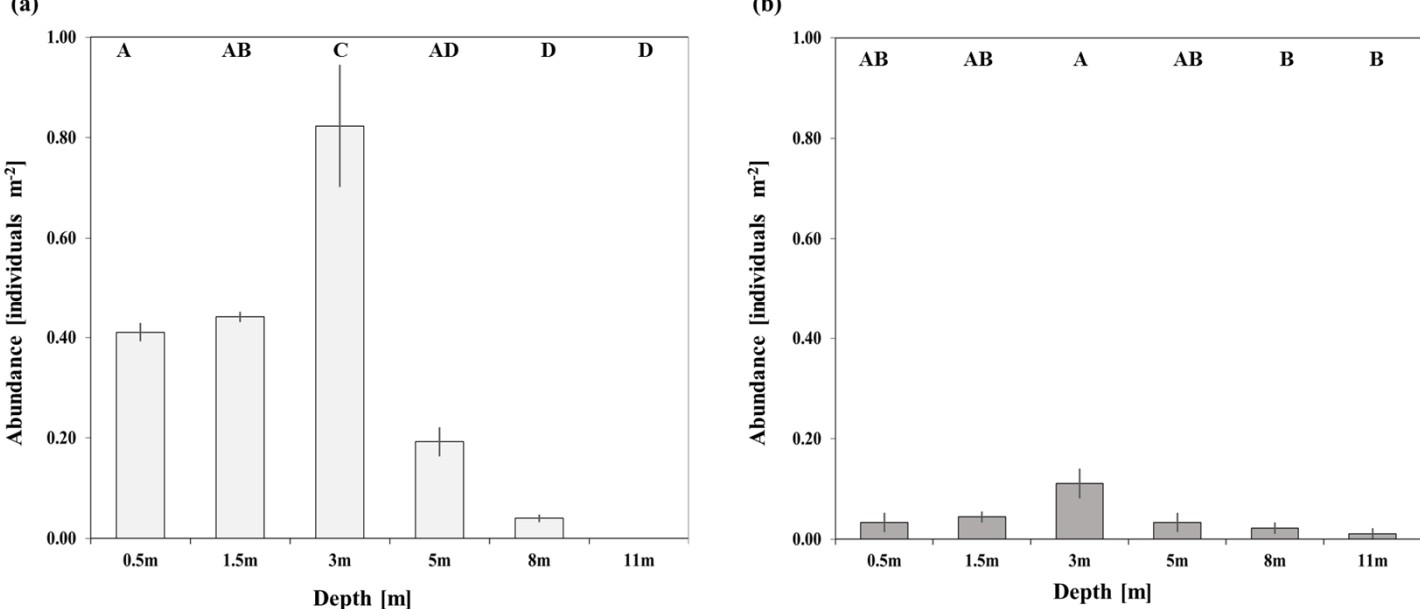

Figure 2: Changes in the abundance [individuals m$^{-2}$ ± SD] of *T. maxima* with depth at a sheltered reef (a) and an exposed reef (b) in the central Red Sea. Different capital letters describe statistically significant differences in abundance between survey depths.





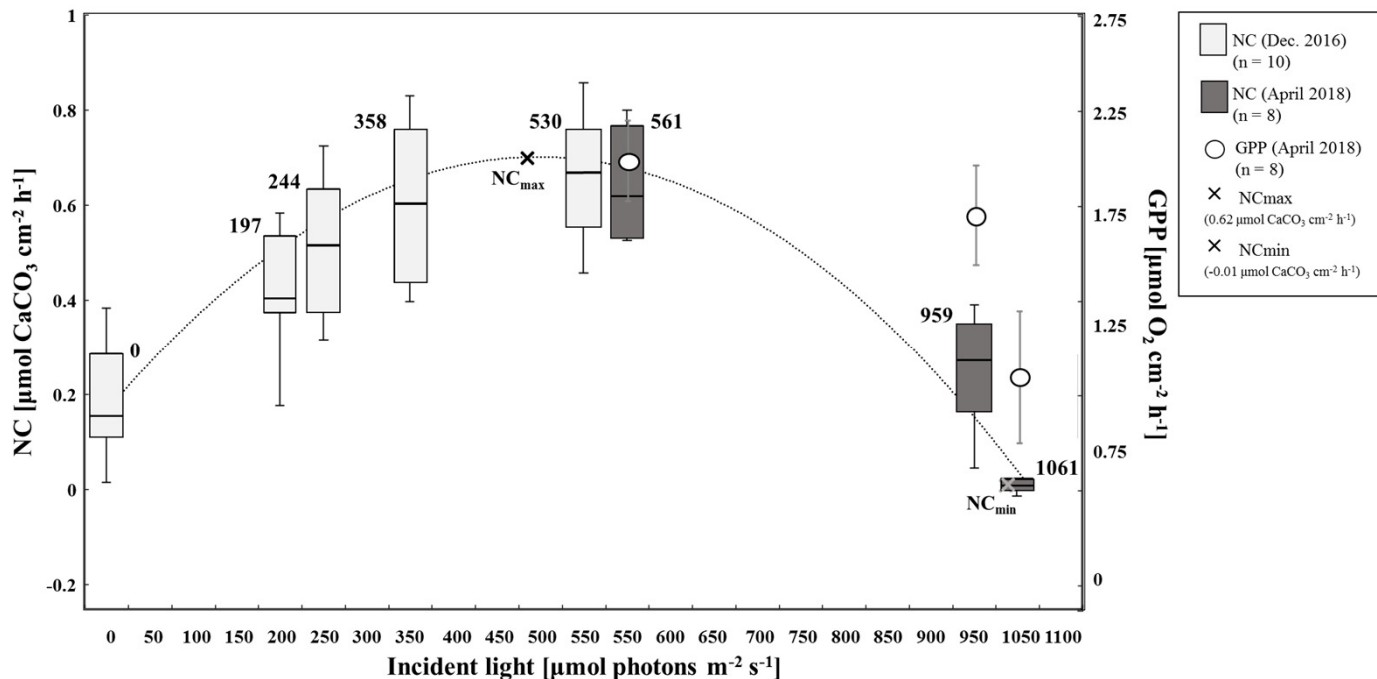

**Figure 3: Box plots showing net calcification rates [µmol CaCO$_3$ cm$^{-2}$ h$^{-1}$] of *T. maxima* under seven different light regimes (197, 244, 358, 530, 561, 959 and 1061 µmol quanta m$^{-2}$ s$^{-1}$) (n = 10 in December 2016 and n = 8 in April 2018) and in the dark, as well as gross primary production [µmol O$_2$ cm$^{-2}$ h$^{-1}$] (n = 8) as dots ($\pm$SE), under three high light regimes (561, 959 and 1061 µmol quanta m$^{-2}$ s$^{-1}$). Calculated maximum net calcification (NC$_{max}$) at 475 µmol quanta m$^{-2}$ s$^{-1}$ and incident light level where dissolution outweighs calcification processes (NC$_{min}$) are symbolized by a cross $\times$. Net calcification rates obtained during incubations under moderate light conditions are symbolised by light grey boxplots, those from the high light incubations by dark grey boxplots, the central line represents the median, the boxes encompass the central 50% of the data and the lines extend to the 95% quartiles.**





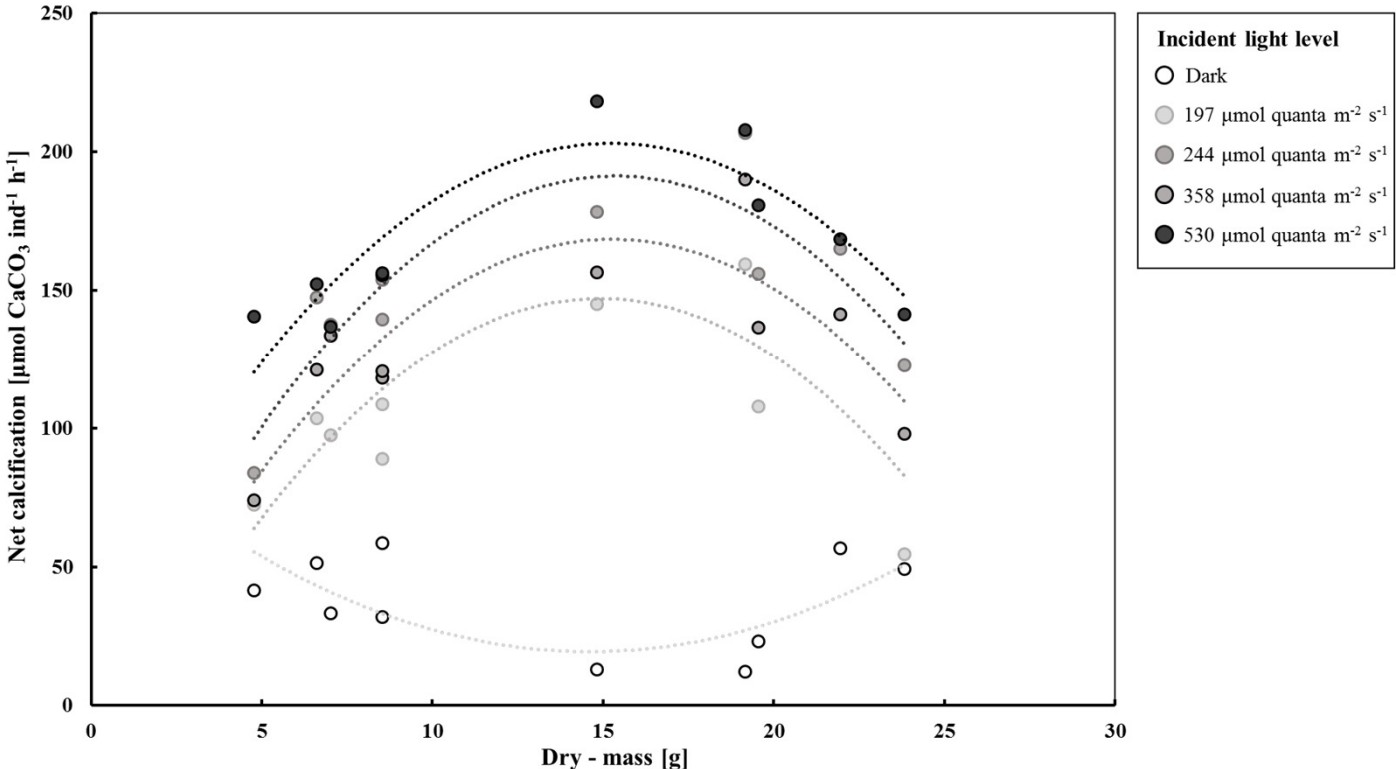

**Figure 4: Net calcification (μmol CaCO$_3$ ind$^{-1}$ h$^{-1}$) (n = 10) in _T. maxima_ at four different incident light levels (197, 244, 358 and 531 μmol photons cm$^{-2}$ h$^{-1}$) and in the dark, plotted against tissue dry-mass (g). Data are shown with polynomial trendlines.**





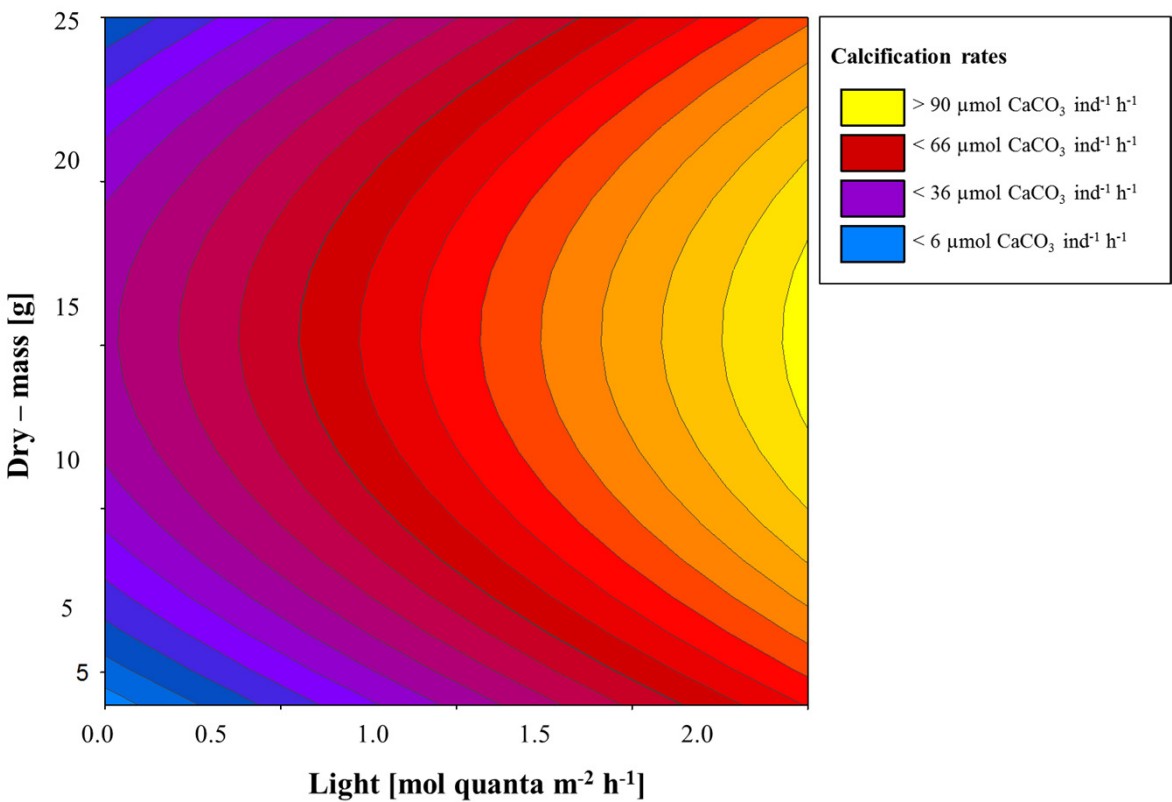

**Figure 5: Model considering a linear relationship between light and calcification and a first order polynomial relationship between dry-mass (DM) and net calcification (NC) explaining 77 % of the variance.**



## Tables

**Table 1:** Net calcification [μmol CaCO$_3$ cm$^{-2}$ h$^{-1}$; ± SE] and gross primary production [μmol O$_2$ cm$^{-2}$ h$^{-1}$; ± SE] under the seven experimental incident light level [μmol quanta cm$^{-2}$ h$^{-1}$] and during the dark.

| Incident light [μmol quanta cm$^{-2}$ h$^{-1}$] | Net calcification [μmol CaCO$_3$ cm$^{-2}$ h$^{-1}$] | Gross primary production [μmol O$_2$ cm$^{-2}$ h$^{-1}$] |
|---|---|---|
| 0 | 0.18 ± 0.02 | - |
| 197 | 0.43 ± 0.04 | N/A |
| 244 | 0.51 ± 0.04 | N/A |
| 358 | 0.60 ± 0.04 | N/A |
| 530 | 0.66 ± 0.05 | N/A |
| 561 | 0.65 ± 0.04 | 2.06 ± 0.24 |
| 959 | 0.25 ± 0.04 | 1.76 ± 0.28 |
| 1061 | 0.01 ± 0.01 | 0.87 ± 0.37 |

5    **Table 2:** Description of the statistical model parameters (Fig.5) combining the influence of irradiance (E, μmol m$^{-2}$ s$^{-1}$) and clam dry-mass (DM; g) on calcification (G; μmol CaCO$_3$ ind$^{-1}$ h$^{-1}$), where d is the fitted intercept:

| $G = a \cdot E + b \cdot DM^2 + c \cdot DM + d$ | | | | | | |
|---|---|---|---|---|---|---|
| | Estimate | SE | t – value df = 46 | p - value | Lower confidence limit [95% CI] | Upper confidence limit [95% CI] |
| a | 0.126 | 0.011 | 11.420 | <0.0001 | 0.104 | 0.148 |
| b | -0.298 | 0.070 | -4.242 | <0.0001 | -0.439 | -0.156 |
| c | 9.115 | 2.006 | 4.545 | <0.0001 | 5.079 | 13.152 |
| d | -29.085 | 11.955 | -2.433 | 0.019 | -53.148 | -5.022 |
| R$^2$: 0.77 | | | | | | |





**Table 3:** Comparison of net calcification rates in relation to light conditions in different marine phototrophic and mixotrophic calcifiers. Values are given as average value mean (± SE) or ᵃ (± SD). Experimental light incubation levels are given in μmol photons m⁻² s⁻¹. Net calcification values were converted to μmol CaCO₃ cm⁻² h⁻¹ from: ᵇ mg CaCO₃ cm⁻² d⁻¹.

| Organism | Species | Nutrition | Region | Light incubation [μmol photons m⁻² s⁻¹] | Net calcification [μmol CaCO₃ cm⁻² h⁻¹] | Method | Study |
|---|---|---|---|---|---|---|---|
| Coral | *Acropora variabilis* | mixotroph | Northern Red Sea | 800 | 0.1 ᵃ | TA anomaly method | **Cohen et al. 2016** |
| Coral | *Porites lutea* | mixotroph | Northern Red Sea | 800 | 0.28 ᵃ | TA anomaly method | **Cohen et al. 2016** |
| Coral | *Porites spp.* | mixotroph | Japan | 700 | 0.79 ᵇ | Buoyant weighing | **Comeau et al. 2014b** |
| Coral | *Pocillophora damicornis* | mixotroph | Japan | 700 | 0.49 ᵇ | Buoyant weighing | **Comeau et al. 2014b** |
| Coral | *Porites compressa* | mixotroph | Hawaii | 698 | 0.81 ± 0.02 ᵇ | Buoyant weighing | **Marubini et al. 2001** |
| Coral | *Acropora pulchra* | mixotroph | French Polynesia | 640 ± 30 | 0.42 ± 0.02 ᵇ | Buoyant weighing | **Comeau et al. 2014a** |
| Coral | *Madracis auretenra* | mixotroph | Caribbean | 200 | 0.36 ± 0.4 ᵃ | TA anomaly method | **Jury et al. 2010** |
| Coral | *Porites compressa* | mixotroph | Hawaii | 150 | 0.43 ± 0.03 ᵇ | Buoyant weighing | **Marubini et al. 2001** |
| Coral | *Acropora pulchra* | mixotroph | French Polynesia | 149.2 ± 0.1 | 0.11 ± 0.01 ᵇ | Buoyant weighing | **Comeau et al. 2014a** |
| **Average Net calcification corals** | | | | | **0.42 ± 0.08** | | |
| Algae | *Porolithion onkodes* | phototroph | Hawaii | 700 | 0.80 ᵃᵇ | Buoyant weighing | **Comeau et al. 2014b** |
| Algae | *Hydrolithion reinboldii* | phototroph | French Polynesia | 640 ± 30 | 0.05 ± 0.00 ᵇ | Buoyant weighing | **Comeau et al. 2014b** |
| **Average Net calcification calcifying algae** | | | | 0.25 ± 0.11 | **0.43 ± 0.38** | | |
| Mollusk | *Tridacna maxima* | mixotroph | Central Red Sea | 1061 | 0.01 ± 0.01 | TA anomaly method | **This study** |
| Mollusk | *Tridacna maxima* | mixotroph | Central Red Sea | 959 | 0.25 ± 0.04 | TA anomaly method | **This study** |
| Mollusk | *Tridacna maxima* | mixotroph | Central Red Sea | 530 – 561 | 0.65 ± 0.03 | TA anomaly method | **This study** |
| Mollusk | *Tridacna maxima* | mixotroph | Central Red Sea | 358 | 0.60 ± 0.04 | TA anomaly method | **This study** |
| Mollusk | *Tridacna maxima* | mixotroph | Central Red Sea | 244 | 0.51 ± 0.04 | TA anomaly method | **This study** |
| Mollusk | *Tridacna maxima* | mixotroph | Central Red Sea | 197 | 0.43 ± 0.02 | TA anomaly method | **This study** |
| **Average Net calcification *T. maxima*** | | | | | **0.47 ± 0.03** | | |

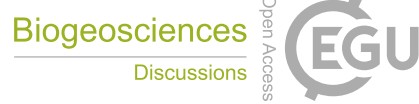



**Table 4:** Comparison of net calcification rates of different marine, heterotrophic calcifiers.
Values are given as average value mean (± SE) or [a] (± SD). All net calcification values were normalized for gram dry-mass (DM), rates given in fresh-weight were converted to DM [b] after Ricciardi and Bourget (1998) or [c] Dame (1972). Net calcification values were converted to $\mu mol\ CaCO_3\ g\ DM^{-1}\ h^{-1}$ from d $\mu g\ CaCO_3\ g\ DM^{-1}\ d^{-1}$.

| Organism | Species | Nutrition | Region | Net calcification [$\mu mol\ CaCO_3\ g\ DM^{-1}\ h^{-1}$] | Method | Study |
|---|---|---|---|---|---|---|
| Coral | *Lophelia pertusa* | heterotroph | North Atlantic | 1.5 [a] | TA anomaly method | **Hennige et al. 2014** |
| Coral | *Madrepora oculata* | heterotroph | Mediterranean Sea | 0.091 ± 0.027 | TA anomaly method | **Maier et al. 2016** |
| **Averaged Net calcification for heterotrophic corals** | | | | **0.80 ± 0.70** | | |
| Mollusk | *Mytilus edulis* | heterotroph | North Sea | 0.0244 [ab] | TA anomaly method | **Gazeau et al. 2007** |
| Mollusk | *Crassostrea gigas* | heterotroph | North Sea | 0.219 [ab] | TA anomaly method | **Gazeau et al. 2007** |
| Mollusk | *Argopecten purpuratus* | heterotroph | Southern Pacific | 0.004 ± 0.001 [d] | Buoyant weighing | **Ramajo et al. 2016** |
| **Averaged Net calcification for heterotrophic mollusks** | | | | **0.08 ± 0.07** | | |
| Mollusk | *Tridacna maxima* | mixotroph | Central Red Sea | 5.38 ± 0.42 | TA anomaly method | **This study** |
| **Averaged Net calcification in *T. maxima*** | | | | **5.38 ± 0.42** | | |