# Peer review of "Light-dependent calcification in Red Sea giant clam Tridacna maxima"

_Biogeosciences, 2018_

## Referee Comment (RC1) · Anonymous Referee #2 · 18 Apr 2019

Rossbach et al. demonstrated depth-dependent abundances of Tridacna maxima in natural reefs and experimentally examined short term net calcification rates of T. maxima in different light conditions. Tridacna is abundant bivalves in coral reefs and has demand as fishery resources and environmental proxies. However, the knowledge about their calcification rates are scarce. While calcification rates of tridacna shells seem to be also strongly related to temperature conditions (Warter et al., 2018), this study provide new insight of the relationship between their calcification and light. I recommend this paper published in "Biogeosciences" after some revisions.

I hope my comments below will be useful to improve the manuscript.

P.3/L7: please check reference style. Probably you can write like "Ip et al., 2006, 2015, 2017".

[Figure]

P.5/L28: …following Dickson et al. (2007).

2.1. Clam abundance surveys: How many belt transects were conducted in each depth?

2.2. Clam incubations: How many clams incubate in each condition?

P.2/L32: I think, after the flow-through system turned off, the incubation tanks should be completely closed to measure carbonate chemistry. This description is needed here. And, how did you sample seawater during the experiments?

P.8/L2: Please refer to "Fig.2" here.

P.10/L3 "In the Red Sea, T. maxima shows a significant increase in net calcification rates with increasing incident light." In your results, strong light conditions over 900 $\mu$mol photons m-2 s-1 made decreasing net calcification rate. So, net calcification rates were not always increasing with light. Is it right?

4.1. Depth-dependent abundances: I think that local geomorphological feature can also change light availability of benthic habitats. Even at same depth, the angle of incident light and local topography makes different shade conditions for each clam. In connection with the matter, please add the detail description of geomorphology at each site in 2.1 and Figure 1.

4.2./P.11/L19: Not only photosynthetic activity, but also the efficiency of photosynthesis and the density of symbionts might intervene between light availability and calcification. Increased light could be also stressor for zooxanthellae (e.g. Weis, 2018). Additional discussion about the influence of light to algal-tridacna holobiont and its calcification processes could persuade the readers of the results in this study.

Fig.1 (b) and (c): Please zoom up the map and point the area of study sites to see topographical differences among two reefs.

Fig.2 and Fig.3: How many specimens did you use for each condition?

[Figure]

S2.1 and S2.2: Legends for each parameter are needed. I couldn't clealy understand the meaning of this table.

Table S2.2.2: Why are the values in the column of "diff" all zero?

[Figure]

---

## Referee Comment (RC2) · Anonymous Referee #1 · 26 Apr 2019

The manuscript investigates a common yet little known question on the light-dependency of iconic giant clams. Using experiments, Authors were able to show interesting results that showed congruence to the species' natural depth distribution.

Some explanations were provided regarding their results, but I think there is scope to expand their Discussion (e.g. how symbiont species may play a role in affecting depth distributions and affect calcification). The current manuscript depth peer review, as well as to provide more details on the mechanism of light-induced calcification and why there is a maximum light threshold before calcification rates drop could be further expounded.

Specific comments:

[Figure]

Abstract, Line 8: Tridacninae is the subfamily for giant clams. Please amend the first sentence.

Introduction, Line 15: Suggest to use Neo et al., 2017 - a review of species status, instead of Neo et al., 2015 that looks at ecological roles. Reference citation: Neo ML, CCC Wabnitz, RD Braley, GA Heslinga, C Fauvelot, S Van Wynsberge, S Andréfouët, C Waters, AS-H Tan, ED Gomez, MJ Costello & PA Todd (2017) Chapter 4. Giant clams (Bivalvia: Cardiidae: Tridacninae): A comprehensive update of species and their distribution, current threats and conservation status. In: Hawkins SJ, Evans AJ, Dale AC, Firth LB, Hughes DJ, Smith IP (eds.), Oceanography and Marine Biology: An Annual Review, Volume 55. Pp. 87–388. CRC Press: Boca Raton, FL.

Discussion, Section 4.1: Suggest Authors to look at the following papers to compare how locations of giant clams (sheltered versus exposed sites) can affect distribution. References: Militz TA, J Kinch & PC Southgate (2016) Population Demographics of Tridacna noae (Röding, 1798) in New Ireland, Papua New Guinea. Journal of Shellfish Research 34(2): 329-335. Neo ML, L-L Liu, D Huang & K Soong (2018) Thriving populations with low genetic diversity in giant clam species, Tridacna maxima and T. noae, at Dongsha Atoll, South China Sea. Regional Studies in Marine Science 24: 278–287.

Discussion, Section 4.2: Suggest Authors to refer to LaJeunesse et al., 2018 (Systematic Revision of Symbiodiniaceae Highlights the Antiquity and Diversity of Coral Endosymbionts) and symbiont-related papers on giant clams (e.g. DeBoer et al., 2012; Ikeda et al., 2017; Lim et al., 2019), and make inferences on how symbiont species may affect depth distribution with respect to light. References: DeBoer TS, AC Baker, MV Erdmann, Ambariyanto, PR Jones & PH Barber (2012) Patterns of Symbiodinium distribution in three giant clam species across the biodiverse Bird's Head region of Indonesia. Marine Ecology Progress Series 444: 117-132. Ikeda S, Yamashita H, Kondo S-n, Inoue K, Morishima S-y, Koike K (2017) Zooxanthellal genetic varieties in giant clams are partially determined by species-intrinsic and growth-related characteristics.

[Figure]

PLoS ONE 12(2): e0172285. Lim SSQ, D Huang, K Soong & ML Neo (2019) Diversity of endosymbiotic Symbiodiniaceae in giant clams at Dongsha Atoll, northern South China Sea. Symbiosis.

---

## Author Comment (AC1) · 20 May 2019

"The manuscript investigates a common yet little known question on the light dependency of iconic giant clams. Using experiments, Authors were able to show interesting results that showed congruence to the species' natural depth distribution. Some explanations were provided regarding their results, but I think there is scope to expand their Discussion (e.g. how symbiont species may play a role in affecting depth distributions and affect calcification). The current manuscript depth peer review, as well as to provide more details on the mechanism of light-induced calcification and why there is a maximum light threshold before calcification rates drop could be further expounded."

We thank the reviewer for the comments, which were very constructive and helpful.

Below, we provide a "Response to reviewer" document detailing, point-by-point, the actions taken to address each comment. The original comments are marked with quotation signs "", and our responses refer to line numbers of the revised manuscript whenever possible, so that the changes can be easily assessed.

Specific comments:

1) "Abstract, Line 8: Tridacninae is the subfamily for giant clams. Please amend the first sentence."

Answer: Abstract, Line 8 was changed to: 'Tropical giant clams of the subfamily Tridacninae, including the species Tridacna maxima, are unique among bivalves as they live in a symbiotic relationship with unicellular algae and generally function as net photoautotrophic.'

2)"Introduction, Line 15: Suggest to use Neo et al., 2017 - a review of species status, instead of Neo et al., 2015 that looks at ecological roles. Reference citation: Neo ML, CCC Wabnitz, RD Braley, GA Heslinga, C Fauvelot, S Van Wynsberge, S Andréfouët, C Waters, AS-H Tan, ED Gomez, MJ Costello & PA Todd (2017) Chapter 4. Giant clams (Bivalvia: Cardiidae: Tridacninae): A comprehensive update of species and their distribution, current threats and conservation status. In: Hawkins SJ, Evans AJ, Dale AC, Firth LB, Hughes DJ, Smith IP (eds.), Oceanography and Marine Biology: AnAnnual Review, Volume 55. Pp. 87–388. CRC Press: Boca Raton, FL."

Answer: We agree with the reviewer and changed the text accordingly:

Page 1, Line 13-16: 'Currently, all giant clam species are listed in the IUCN Red List of Threatened Species (IUCN, 2016) and protected under Appendix II of the Convention on International Trade in Endangered Species of Wild Fauna and Flora (CITES). Most of them are considered under a lower risk / conservation dependent status, however the IUCN status of tridacnine species, is in need of updating according to Neo et al. (2017).

3) "Discussion, Section 4.1: Suggest Authors to look at the following papers to compare how locations of giant clams (sheltered versus exposed sites) can affect distribution.

References: Militz TA, J Kinch & PC Southgate (2016) Population Demographics of Tridacna noae (Röding, 1798) in New Ireland, Papua New Guinea. Journal of Shellfish Research 34(2): 329-335. Neo ML, L-L Liu, D Huang & K Soong (2018) Thriving populations with low genetic diversity in giant clam species, Tridacna maxima and T. noae, at Dongsha Atoll, South China Sea. Regional Studies in Marine Science 24: 278–287."

Answer: We thank the reviewer for this comment and added the following paragraph to the discussion:

Page 10, Line 17-33:

'Explanations for the observed contrasts in numbers of clams per m2 at both reefs could lay in the probable differences in abiotic environmental conditions at the surveyed sites. For instances, giant clams at the exposed reef are potentially more at risk from high wave action than at the sheltered reef site, which could impact the initial settlement (Jameson, 1976) as well as the survival of juveniles (Foyle et al., 1997), as both have been shown to be influenced by geographical factors (Foyle et al., 1997). While a previous study (Militz et al., 2015), in which abundances of giant clam species in French Polynesia were examined, report similar patterns for T. crocea, opposite patterns were observed for abundances of T. maxima in that region. In the reefs surveyed by Militz and collegues (2015), T. maxima showed higher abundances at reef sites with a high exposure, in comparison to those with low exposure levels. However, additional factors such as temperature and local geomorphology might also have an influence on giant clam densities. Therefore, it is not possible to confidently identify the underlying causes for the observed differences by considering exposure alone. For example, T. maxima specimens from our study, which were located at the more southern reef, could be possibly also exposed to higher surface water temperatures due to location

of this reef at lower latitudes. Mean seasurface annual temperature of the Red Sea have been shown to increase towards lower latitudes and can be as high as 33 °C in the Central and Southern Red Sea (Chaidez et al., 2017). Further, the local geomorphological features of each reef could influence the light availability of benthic habitats. Consequently, differences in the local topography could have led to different angles of incident light and shading conditions, which would then result in differences between reefs even though the examined depths are identical.

4) "Discussion, Section 4.2: Suggest Authors to refer to LaJeunesse et al., 2018 (Systematic Revision of Symbiodiniaceae Highlights the Antiquity and Diversity of Coral Endosymbionts) and symbiont-related papers on giant clam(e.g. DeBoer et al., 2012; Ikeda et al., 2017; Lim et al., 2019), and make inferences on how symbiont species may affect depth distribution with respect to light.

References: DeBoer TS, AC Baker, MV Erdmann, Ambariyanto, PR Jones & PH Barber (2012) Patterns of Symbiodinium distribution in three giant clam species across the biodiverse Bird's Head region of Indonesia. Marine Ecology Progress Series 444: 117-132. Ikeda S, Yamashita H, Kondo S-n, Inoue K, Morishima S-y, Koike K (2017) Zooxanthellal genetic varieties in giant clams are partially determined by species-intrinsic and growth-related characteristics.PLoS ONE 12(2): e0172285. Lim SSQ, D Huang, K Soong & ML Neo (2019) Diversityof endosymbiotic Symbiodiniaceae in giant clams at Dongsha Atoll, northern South China Sea. Symbiosis."

Answer: We thank the reviewer for this valuable comment and added the following paragraph to the manuscript:

Page 12, Line 11-25: 'Giant clams, including T. maxima can potentially harbour multiple genera of Symbiodiniaceae simultaneously (DeBoer et al., 2012;Ikeda et al., 2017), including Symbiodinium, Cladocopium and Durusdinium (previously referred to as clade A, C and D (LaJeunesse et al., 2018)) (DeBoer et al., 2012). The composition of these associated algal symbionts might therefore also impact the susceptibility to (high) light

levels, as different genera of Symbiodiniaceae (in symbiosis) exhibit different physiological and ecological patterns, including sensitivity to light and temperature (Rowan et al., 1997;Berkelmans and Van Oppen, 2006). However, a previous study on Red Sea giant clams and their associated Symbiodiniaceae (Pappas et al., 2017), report that T. maxima in the region exclusively associated with Symbiodinium spp. (previously clade A) which was thus assumed to represent an optimal group for the local environmental conditions. Yet, the reliance of calcification of host organisms (e.g. T. maxima) on their relationship with symbiotic algae could provide an explanation for the significant decrease in net calcification rates at the highest light treatment (1061 $\mu$mol photons m-2 s-1). These diminished rates could be the result of photoinhibition and even photodamage of the associated unicellular algae, when exposed to these high incident light levels. This would be also supported by the pronounced decrease in gross primary production rates at this light treatment. High incident light level, especially high level of UV radiation in shallow waters, have been previously shown to be correlated with decreased calcification rates in other marine calcifiers such as stony corals, e.g. Porites compressa (Kuffner, 2001).

---

## Author Comment (AC2) · 20 May 2019

We thank the reviewer for the comments, which were very constructive and helpful. Below, we provide a "Response to reviewer" document detailing, point-by-point, the actions taken to address each comment. The original comments are marked with quotation signs "", with our responses refering to line numbers of the revised manuscript whenever possible, so that the changes can be easily assessed.

1) "Rossbach et al. demonstrated depth-dependent abundances of Tridacna maxima in natural reefs and experimentally examined short term net calcification rates of T. maxima in different light conditions. Tridacna is abundant bivalves in coral reefs and has demand as fishery resources and environmental proxies. However, the knowledge

about their calcification rates are scarce. While calcification rates of tridacna shells seem to be also strongly related to temperature conditions (Warter et al., 2018), this study provide new insight of the relationship between their calcification and light. I recommend this paper published in "Biogeosciences" after some revisions.

I hope my comments below will be useful to improve the manuscript. We thank the Reviewer for the comments, they were very constructive and helpful. The journal shared the recommendations from the reviewer previously (as a quick report) and we amended the manuscript following the reviewer's suggestions before its publication online in Biogeosciences Discussions. We provide a "Response to reviewer" document detailing, point-by-point, the actions taken to address each comment. The original comments are shown in black with our responses in blue and refer to line numbers of the revised manuscript whenever possible, so that the changes can be easily assessed."

2) "P.3/L7: please check reference style. Probably you can write like "Ip et al., 2006, 2015, 2017"."

Answer: The references have been changed accordingly.

3) "P.5/L28: . . .following Dickson et al. (2007)."

Answer: Changed to 'following Dickson et al. (2007).'

4) "2.1. Clam abundance surveys: How many belt transects were conducted in each depth?"

Answer: At the sheltered reef, we conducted six transects at each depth, at the exposed reef three at each depth. We added this information to the manuscript which now reads:

Page 4, Line 8 -10: 'At the sheltered reef, a total area of 1,000 m2 was covered and we conducted six transects at each depth. At the exposed reef 560 m2 were covered, with three transects at each depth.'

5) "2.2. Clam incubations: How many clams incubate in each condition?"

Answer: For the incubations at moderate light levels (530, 358, 244 and 197 $\mu$mol quanta m-2 s-1) we used 20 clams which were set in pairs (2 clams each) in the ten, independent incubation chambers (e.g. ten replicates). Net calcification was then later normalized for mantle surface area (cm2) or tissue dry-mass (g) of the clams. Each pair (n=10) was incubated at all four light levels. For the incubations at high light levels (561, 959 and 1061 $\mu$mol quanta m-2 s-1), 8 clams were set in individual incubations chambers (e.g. 8 replicates). All 8 clams were incubated at 561 and 959 $\mu$mol quanta m-2 s-1, while at 1061 $\mu$mol quanta m-2 s-1 only six clams were incubated as two died after the incubations at the second highest treatment.

The following has been changed in the manuscript:

Page 4, Line 22-23: 'The experimental setup consisted of ten flow-through independent LDPE (low density polyethylene) outdoor aquaria (30 L). Each aquaria contained two clams (in total 20), cleaned with a brush from epibionts prior to the experiment.' Page 4, Line 31-33: 'We conducted short-term incubations of 6 hours (from approx. 09:30 to 15:30 mean solar time) under four different shadings and one dark incubation (at night) (n = 10), allowing 3 days acclimatization period to the clams, prior to each incubation.'

6) "P.2/L32: I think, after the flow-through system turned off, the incubation tanks should be completely closed to measure carbonate chemistry. This description is needed here."

Answer: The aim of the first set of incubations (moderate light level) was to measure the calcification rate of the clams using the TA anomaly method. We are providing DIC, omega and the rest of the carbonate system only to show that the incubation did not generated an artifact of extreme low or high omega aragonite due to TA reduction, combined with net DIC emission or uptake by photosynthesis and respiration. It is right that in the event that we would have wanted to evaluate primary production / respiration rate through the variation of DIC, the tanks should have been hermetically closed to

avoid air-sea CO2 exchange. However, our goal was to maintain ambient pCO2 in our incubations by allowing air-water gas exchange, thereby preventing extreme variations of omega that would have affected the calcification rates.

7) "And, how did you sample seawater during the experiments?"

Answer: Seawater was sampled using gas tight 100 mL borosilicate bottles.

Changes:

Page 5, Line 28-29: 'At the start, after three and after six hours of incubation, seawater was sampled from each experimental aquaria in gas tight 100 mL borosilicate bottles (Schott Duran, Germany) and poisoned with mercury chloride, following Dickson et al., 2007.'

8)" P.8/L2: Please refer to "Fig.2" here."

Answer: We added the reference to Figure 2.

9) "P.10/L3 "In the Red Sea, T. maxima shows a significant increase in net calcification rates with increasing incident light."  In your results, strong light conditions over 900 $\mu$mol photons m-2 s-1 made decreasing net calcification rate.  So, net calcification rates were not always increasing with light. Is it right?"

Answer: Yes, we agree.  In fact, net calcification rates show a significant dependence of incident light levels. We changed the sentence and it now reads:

Page 10, Line 3: 'In the Red Sea, T. maxima shows a significant dependence of net calcification rates with incident light.'

10)" 4.1.  Depth-dependent abundances: I think that local geomorphological feature can also change light availability of benthic habitats.  Even at same depth, the angle of incident light and local topography makes different shade conditions for each clam. In connection with the matter, please add the detail description of geomorphology at each site in 2.1 and Figure 1."

Answer: We thank the reviewer for this comment and agree that the local geomorphology of the reefs could influence the incident light levels as well. Unfortunately, we did not conduct a more detailed survey on the overall topography of the observed reefs. Thus, the only information that can be shared is what is already part of the manuscript: Page 4, Line 4-7: 'The first station was Abu Shosha (22.303833 N, 39.048278 E), a small inshore reef, were abundances were examined at the sheltered, leeward side (Southeast) of the reef, which are relatively protected from wave action and currents (Khalil et al., 2013). Additionally, abundances were assessed at a second station (20.753764 N, 39.442561 E), a fringing reef close to Almojermah, were we conducted transects at the exposed, windward side (Northwest) of the reef.'

However, we added the following sentence to the discussion:

Page 10, Line 17-33 'Explanations for the observed contrasts in numbers of clams per m2 at both reefs could lay in the probable differences in abiotic environmental conditions at the surveyed sites. For instances, giant clams at the exposed reef are potentially more at risk from high wave action than at the sheltered reef site, which could impact the initial settlement (Jameson, 1976) as well as the survival of juveniles (Foyle et al., 1997), as both have been shown to be influenced by geographical factors (Foyle et al., 1997). While a previous study (Militz et al., 2015), in which abundances of giant clam species in French Polynesia were examined, report similar patterns for T. crocea, opposite patterns were observed for abundances of T. maxima in that region. In the reefs surveyed by Militz and collegues (2015), T. maxima showed higher abundances at reef sites with a high exposure, in comparison to those with low exposure levels. However, additional factors such as temperature and local geomorphology might also have an influence on giant clam densities. Therefore, it is not possible to confidently identify the underlying causes for the observed differences by considering exposure alone. For example, T. maxima specimens from our study, which were located at the more southern reef, could be possibly also exposed to higher surface water temperatures due to location of this reef at lower latitudes. Mean seasurface annual temperature of

the Red Sea have been shown to increase towards lower latitudes and can be as high as 33 °C in the Central and Southern Red Sea (Chaidez et al., 2017). Further, the local geomorphological features of each reef could influence the light availability of benthic habitats. Consequently, differences in the local topography could have led to different angles of incident light and shading conditions, which would then result in differences between reefs even though the examined depths are identical.

A more detailed satellite photo of the reefs is unfortunately not available.

11) "4.2./P.11/L19: Not only photosynthetic activity, but also the efficiency of photo-synthesis and the density of symbionts might intervene between light availability and calcification. Increased light could be also stressor for zooxanthellae (e.g. Weis, 2018). Additional discussion about the influence of light to algal-tridacna holobiont and its cal-cification processes could persuade the readers of the results in this study."

Answer: Agreed. Thus we added the following paragraph to the discussion:

Page 12, Line 3-9: 'The reliance of calcification of calcifying host organism (e.g. T. maxima) on their relationship with symbiotic algae could also provide an explanation for the significant decrease in net calcification rates at the highest light treatment (1061 $\mu$mol photons m-2 s-1). These diminished rates could be the result of photoinhibition and even photodamage of the associated Symbiodinium algae, when exposed to these high incident light levels. This would be also supported by the pronounced decrease in gross primary production rates at this light treatment. High incident light level, espe-cially high level of UV radiation in shallow waters, have been previously shown to be correlated with decreased calcification rates in other marine calcifiers such as stony corals, e.g. Porites compressa (Kuffner, 2001).'

12) "Fig.1 (b) and (c): Please zoom up the map and point the area of study sites to see topographical differences among two reefs."

Answer: Please see reply to previous question.

13) "Fig.2 and Fig.3: How many specimens did you use for each condition?"

Answer: We added the number of replicates to the legend of figure 3

and changed the description to:

Figure 3: 'Box plots showing net calcification rates [$\mu$mol CaCO3 cm-2 h-1] of T. maxima under seven different light regimes (197, 244, 358, 530, 561, 959 and 1061 $\mu$mol quanta m-2 s-1) (n = 10 in Dec. 2016 and n = 8 in April 2018) and in the dark as well as gross primary production [$\mu$mol O2 cm-2 h-1] (n = 8) as dots ($\pm$SE), under three high light regimes (561, 959 and 1061 $\mu$mol quanta m-2 s-1).'

As well as the description of Figure 4:

Figure 4: 'Net calcification ($\mu$mol CaCO3 ind-1 h-1) (n = 10) in T. maxima at four different incident light levels (197, 244, 358 and 531 $\mu$mol photons cm-2 h-1) and in the dark, plotted against tissue dry-mass (g). Data are shown with polynomial trendlines.'

14) "S2.1 and S2.2: Legends for each parameter are needed. I couldn't clealy understand the meaning of this table. "

Answer: We changed the table legends to:

Table S1.1 'Seawater carbonate chemistry conditions at start of incubations under moderate light conditions (530, 358, 244, 197 $\mu$mol photons m-2 s-1) and during the dark. Total alkalinity (TA) and dissolved inorganic carbon (DIC) were measured, while the inorganic carbon speciation , including pH, partial pressure of carbon dioxide (pCO2), carbon dioxideaq (CO2(aq)), bicarbonate (HCO3-), carbonate (CO32-) as well as the aragonite ($\Omega$Arag) and calcite ($\Omega$Calc) saturation state were calculated using R package Seacarb. Values are means $\pm$ SD (n = 10).'

Table S1.2 'Total alkalinity (TA) at start of each incubation under high light conditions (1061, 959, 561 $\mu$mol photons m-2 s-1) and during the dark. Values are means $\pm$ SD (n = 8).'

15) "Table S2.2.2: Why are the values in the column of "diff" all zero?"

Answer: The values were < 0.0001 and since we only reported them up to the 4th decimal place they appear in the table as 0. Respective values in the table are now changed to < 0.0001

———————————————————

[Figure]

**Figure 3:** Box plots showing net calcification rates [μmol CaCO₃ cm⁻² h⁻¹] of *T. maxima* under seven different light regimes (197, 244, 358, 530, 561, 959 and 1061 μmol quanta m⁻² s⁻¹) (n = 10 in December 2016 and n = 8 in April 2018) and in the dark, as well as gross primary production [μmol O₂ cm⁻² h⁻¹] (n = 8) as dots (±SE), under three high light regimes (561, 959 and 1061 μmol quanta m⁻² s⁻¹). Calculated maximum net calcification (NC$_{max}$) at 475 μmol quanta m⁻² s⁻¹ and incident light level where dissolution outweighs calcification processes (NC$_{min}$) are symbolized by a cross ×. Net calcification rates obtained during incubations under moderate light conditions are symbolised by light grey boxplots, those from the high light incubations by dark grey boxplots, the central line represents the median, the boxes encompass the central 50% of the data and the lines extend to the 95% quartiles.

**Fig. 1.** Figure 3